# Alcohol Selective Optical Sensor Based on Porous Cholesteric Liquid Crystal Polymer Networks

**DOI:** 10.3390/molecules27030773

**Published:** 2022-01-25

**Authors:** Tai-Yuan Yeh, Ming-Fu Liu, Ru-De Lin, Shug-June Hwang

**Affiliations:** Department of Electro-Optical Engineering, National United University, Miao-Li 360, Taiwan; taiyuan.ee10@nycu.edu.tw (T.-Y.Y.); zx74361296@gmail.com (M.-F.L.); xup6aul4eji3@gmail.com (R.-D.L.)

**Keywords:** cholesteric liquid crystal polymer, alcohol sensor, methanol detection, carboxylate polymer

## Abstract

A responsive hydrogen-bonded cholesteric liquid crystal polymer (CLCP) film with controlled porosity was fabricated as an optical sensor to distinguish between methanol and ethanol in alcohol solutions. To facilitate responding the alcohols, porosity was generated by removing the nonreactive liquid crystal agent, and the hydrogen bridges of CLCP were broken. The sensitivities of CLCPs to ethanol and methanol were obtained by monitoring the wavelength shifts of the transmission spectrum at different alcohol concentrations and ratios of methanol/ethanol. Changes in the central wavelength of the CLCP network transmission spectrum allowed the methanol–ethanol ratio to be discriminated. A linear relationship between wavelength shift of CLCP networks and alcohol concentration was obtained experimentally, and the sensor characteristics were explored. The sensitivities of the CLCPs were 1.35 and 0.18 nm/% to ethanol and methanol, respectively. The sensing sensitivity of cholesteric networks to alcohol molecules increased as the methanol–ethanol ratio declined. Therefore, CLCP could act as a stimuli-responsive material to distinguish the concentrations of acetone and ethanol in mixed solutions. Furthermore, the impact of UV intensity for curing a CLC mixture on the sensing sensitivity to the different alcohol concentrations was also studied. The higher UV intensity could enhance the sensitivity to alcohol molecules and distinguishing ability between methanol and ethanol.

## 1. Introduction

Methanol and ethanol are colorless, flammable, and easily soluble in water, and are commonly used in medical sterilization, chemical synthesis, and industrial fuels. Since these two alcohol solutions are easy to obtain and the price of methanol is lower than that of ethanol, some unscrupulous manufacturers maliciously make alcoholic beverages mixed with methanol. However, if methanol is ingested, inhaled, or absorbed by the skin, it can cause irreversible tissue injury, especially to the eyes and nervous system, and even death. To effectively prevent the threat of methanol to human health, the illegal use of methanol as fake ethanol in alcoholic beverages must be detected easily and reliably. However, the high similarity of methanol and ethanol makes methanol sensing surprisingly challenging.

Various methods to distinguish between methanol and ethanol have been reported, such as gas chromatography-mass spectrometry [1], spectrofluorometric [2], optical fiber sensing [3,4,5], high performance liquid chromatography (HPLC) [6], electrochemical methods [7], Raman spectroscopy [8], and multifunctional nanomaterials [9,10]. However, most of the above methods require cumbersome sample preparation, expensive equipment, and complicated operation, which greatly limits the practicality of the sensor. Therefore, a method allowing simple, convenient, and selective distinction between ethanol and methanol in aqueous solutions is urgently required.

Birefringent liquid crystals (LCs) are sensitive to the influence of environmental fields which change the effective refractive index of LC molecules and alter their optical properties. LC-based biosensors combine the advantages of high sensitivity, good repeatability, and good biocompatibility, so they have recently attracted considerable attention and are extensively used in physical sensing [11,12], chemical sensing [13,14], and biological sensing [15,16]. Cholesteric liquid crystals (CLCs) with natural periodicity can be regarded as one-dimensional photonic bandgap materials with central wavelength $\lambda_{c}=\bar{n}p$ and bandwidth Δλ=Δnp, where $\bar{n}$ is average refractive index, Δn is the birefringence of the LC host, and p is CLC pitch length. The helical structure is susceptible to disturbances caused by external stimuli (e.g., temperature, electric voltage, chemical agents, pH environment), which in turn significantly change the CLC alignment and helical pitch, altering the wavelength of the reflected light [17]. This unique optical response, induced by external stimuli, can be transduced into macroscopic responses; for example, changes in the reflected colors of CLCs that are visible to the naked eye. Therefore, CLCs can act as stimuli-responsive materials for colorimetry, and their potential has been investigated for more and more chemical sensor applications over the last decade [18,19,20,21,22,23,24,25,26].

The unique property of CLCs has attracted great interest to develop visual sensing materials that use color change as a detection signal. To enhance the stability, repeatability, and ease of operation of CLC films, some researchers have proposed a sensor platform based on cholesteric liquid crystal polymer (CLCP) network structure, which can be fabricated easily on a single glass or plastic substrate [14,19,21,22,23,24,25,26]. Owing to the CLCP sensor can be implemented in a solid-state film with simple structure, high mechanical stability, convenient operation, and accurate measurement, it has become a hot research topic in the current sensing field [14,19,21,22,23,24,25,26]. Among these technologies of CLCP, C. K. Chang et al. proposed a new photonic CLC material, which is a hydrogen-bridged CLCP network with porosity, to distinguish methanol from ethanol [19,22]. To improve the expansion degree of the helical structure and enhance the sensing capability, they introduced porosity to the polymer networks. The creation of porosity in the CLCP networks not only facilitates the diffusion of the analytes into the CLC film, but also increases the surface-area-to-volume ratio of the polymer networks that react with the analytes. The hydrogen-bridged CLCP has carboxylic moieties, and its hydrogen bonds can be activated to interact with the hydroxyl group of analytes as treated with alkaline solution. As a result, the bond-breaking treatment stimulates the absorption of alcohol molecules having hydrogen bonds. These CLCP networks with the carboxyl groups having different affinities for methanol and ethanol molecules are very sensitive to alcohol molecules. When the carboxylate polymer absorbs alcohol molecules with different molecular polarities, different degrees of expansion occur, resulting in the CLC film showing different reflected wavelength changes.

Although the previous works have demonstrated the discrimination between methanol and ethanol by using hydrogen-bridged cholesteric polymer networks [19,22], the authors mainly discussed the sensing characteristics of CLCP films fabricated by different CLC mixtures to alcohol molecules. There are some problems that need to be resolved further, such as (1) the optimum activation time of alkaline solution treatment, (2) too long detection time (~10 min), (3) poor reversibility, and so on. In this work, the commercial diacrylate reactive mesogen LC242 was utilized as the host LC material to achieve a cost-effective biosensor for determining ethanol and methanol. LC242 cannot directly interact with alcohol molecules, so to differentiate methanol from ethanol, porous hydrogen-bonded CLC polymer (CLCP) networks were applied, in which the hydrogen bridges of benzoic acid based dimers in the networks were used and needed to be broken and activated using an alkaline solution. Here, to obtain the optimum activation time of CLCP networks for alkaline solution treatment, the transient spectra of CLCP film were monitored during the bond-breaking process. According to the experimental result, the activated time for breaking the hydrogen bond of CLCP was found to be ~130 s. In order to verify the ability of the prepared CLC film to detect alcohol solutions, the transmission spectra were measured using alcohol solutions with different methanol and ethanol mixing ratios. Alcohol absorption was monitored by measuring changes in the transmission band of the cholesteric network due to the increased helical pitch of the CLC when the polymer networks swelled by fast in-diffusion of alcohol molecules. Based on the experimental results, the total alcohol concentration and proportion of methanol and ethanol in solutions could be detected in 30 s by monitoring the reflected wavelength shift of CLCP. As the methanol–ethanol ratio decreased, the sensing sensitivity of the cholesteric networks to alcohol molecules increased.

Additionally, the influence of UV intensity for curing the CLC mixture on the sensing sensitivity was explored. Higher UV intensity was found to enhance sensitivity to alcohol molecules and improve discrimination between methanol and ethanol. Moreover, the thermal treatment at 50 °C for 1 min was also applied to make the alcohol molecules desorbed from the hydrogen-bridged CLC networks for recovering the CLCP films. It has been proved that CLCP networks still preserved good reversibility after five repeated tests. The response of CLCP to alcohols yielded quickly reversible color changes, indicating that hydrogen-bonded CLCP could quickly absorb and release alcohol molecules by thermal treatment. This work demonstrated that CLCP film had good specificity for the detection of alcohol solutions, provided rapid detection with high stability, and offered cost-effective and simple spectral or visual observation. As a first application, a CLCP network sensor for screening alcoholic beverages could help to prevent outbreaks of methanol poisoning, and it is believed that this technology has broad applicability and strong commercial development potential in the future. The CLCP sensor provides an easy to use, low-cost, rapid and repeatable method for the detection and quantification of methanol/ethanol contamination in solutions and shows strong potential as a cost-effective sensor for other liquid sensing applications.

## 2. Results and Discussion

To enable the absorption of alcohol, the hydrogen bonds between 4-(acryloyloxyhexyloxy) benzoic acid (6OBA) and 4-(6-acryloyloxy-n-hex-1-yloxy)-2-methylbenzoic acid (6OBAM), which are the hydrogen-bonding molecular triggers and form the dimer through the hydrogen bond, were broken and activated to form carboxylic salt (COO^–^Na^+^) using an alkaline solution, as shown in Figure 1. The duration of the bond-breaking process considerably affects the performance of the sensor, so to obtain the optimum bond-breaking process, the transient spectrum was recorded.

During the bond-breaking process, the reflected color of CLCP film cured at UV intensity of 20 mW/cm^2^ was observed to gradually change from blue through purple to orange-red. The complete color change took 130 s, as shown in Figure 2a. The reflected color of CLCP film showed a red shift during the activation process, which was the result of swelling of the helical structure of the CLC due to the broken hydrogen bonds in the CLC networks and the accompanying absorption of alkaline solution. Figure 2b shows the time-dependent transmission spectrum of the CLCP film. The transmittance at the peak wavelength near 441 nm was observed to gradually increase with increasing immersion time, and another new transmittance band at peak wavelength near 565 nm was generated after a few seconds (~10 s). The transmittance of this new band decreased with immersion time, and its peak wavelength and bandwidth became progressively red-shifted and broader over time. However, when the immersion time was beyond 130 s, the original band at ~441 nm completely disappeared, and the new band had no obvious wavelength shift over time. Nevertheless, it was also found that too long soaking time can unexpectedly cause the CLCP film to detach from the substrate surface. According to the experimental results, the time required for breaking the hydrogen bond of CLCP was selected as 130 s in this work.

To study the alcohol sensing characteristics of the functionalized CLCP film, 40 μL of 40% ethanol solution was dropped onto the CLC film, and the corresponding dynamic spectral response of CLC was measured as shown in Figure 3a. It demonstrated that the alcohol solution caused the wavelength to shift from green to red within 5 s, and the new transmission band was broader than that of the original CLCP film. The new peak wavelength of the CLC film red-shifted further with time as the number of ethanol molecules absorbed by the CLC film increased. The red-shift phenomenon was due to rapid absorption of ethanol molecules by carboxylic salt groups, resulting in expansion of the polymer networks due to swelling, and a decrease of the order of the CLC helical structure which caused the transmission band to widen. Maximum red shift occurred after ~20 s and was followed by a small blue shift after around 30 s. This unexpected blue shift, with a small accompanying decrease in bandwidth, is highlighted by the blue arrow in Figure 3a and shown in detail in Figure 3b. The transmission band for blue shift was narrower than that for red shift, and the amount of blue shift was minor. The minor blue shift was likely due to the release or desorption from the film of a small fraction of alcohol molecules, leading to shrinkage. Shrinkage of the film resulted in a decrease of the pitch, causing a blue shift of the transmission band and an increase in the order of the CLC helical structure.

Figure 4 shows the time-dependent wavelength shift of the CLC film under different ethanol concentrations. The experimental results showed that the wavelength shift increased rapidly with detection time and reached a maximum at a certain time, as well as more ethanol molecules contained in the alcohol solution caused the CLC film to exhibit a larger red shift. Nonetheless, the wavelength shift decreased slightly with time after the time of wavelength shift reached the maximum value. The tiny decrease at longer times may be because there were no more ethanol molecules interacting with the hydrogen-bridged CLC networks and at the same time, the ethanol molecules in the CLC film were desorbing, so a small amount of wavelength blue shift occurred. These results showed that CLC films were able to quickly produce a visible response that was sensitive to ethanol concentration.

To evaluate the ability of hydrogen-bonded CLCP networks to sense and differentiate ethanol and methanol, solutions were prepared with 60% water and 40% methanol–ethanol mixture, in which the methanol–ethanol ratio was varied. Forty microliters of alcohol solution was dropped onto the CLCP film, and the transmission spectrum was measured 30 s later. Figure 5a shows the spectral response of the CLCP film to the various alcohol solutions. It can be seen the wavelength shifts induced by the uptake of methanol and ethanol molecules were significantly different. This was due to the different molecular affinities of methanol and ethanol with hydrogen-bonded CLC. The molecular affinity of ethanol is larger than that of methanol with hydrogen-bonded CLC polymer networks [19,22], so alcohol solutions containing more ethanol molecules cause the helical structure of the CLC to swell more. Therefore, the central wavelength of the CLCP film showed a larger red shift as the ratio of methanol–ethanol was reduced. Figure 5b shows the wavelength shift of the CLC films with different methanol–ethanol ratios, in which a linear fit between wavelength shift and the alcohol concentration and the ratio of methanol/ethanol was obtained by the least squares method. The slope of the linear fit denotes the selectivity of the CLC sensor between methanol and ethanol for a fixed total alcohol concentration of 40%.

The sensing performance of the CLCP film was determined at different alcohol concentrations from 0% to 40% and different ethanol–methanol ratios. Figure 6 shows that wavelength red shift increased with increasing the alcohol concentration, regardless of the mixture of methanol and ethanol at any ratio. At the same alcohol concentration, the greater the proportion of ethanol, the greater the wavelength shift. This phenomenon was attributed to the molecular affinity of ethanol with a hydrogen-bonded CLCP networks higher than that of methanol, so the contribution of the same amount of ethanol to the wavelength shift was larger than that of methanol. Linear fits are shown for wavelength shift versus total alcohol concentration, and the sensing sensitivity of CLC networks to alcohol molecules increased as the methanol–ethanol ratio declined. The sensitivity for ethanol and methanol was 1.35 and 0.18 nm/%, respectively. Consequently, the CLCP films illustrated good selective responses to methanol from ethanol. Additionally, from the experimental results of Figure 6, it could be also observed the transmission central wavelength of the CLC film immersed in pure water exhibited a red shift to around 210 nm, which was ~48.8% wavelength-increment caused by shrinkage of the helical pitch. The swelling degree of the porous film in alcohol solutions mixed by altered ratio of methanol and ethanol was considerably different, for example, the expansion degree of helical structure induced by 40% ethanol and methanol solution was obtained as ~61.4% and ~50.5%, respectively. Compared with the previous works [19,22], in which the transmission spectrum was recorded after soaking the sample in solutions for 10 min, although the sensitivity of our CLCP films for sensing alcohol molecule was lower, a very small volume (~40 µL) of alcohol solution could be distinguished in a very short time. The detection time (~30 s) was significantly shortened.

The UV curing intensity during the polymerization process affects the size of the voids within the polymer networks [27,28], significantly influencing the response of CLC films to alcohol solutions. Higher curing intensity generally results in more open polymer networks with larger voids. High porosity CLC polymer networks can increase the diffusion rate of alcohol molecules and the expansion of the helical structure of CLC, thereby enhancing the sensitivity to analytes. Therefore, in addition to using nonreactive mesogen 5CB as template molecules to create the porous structure, the microstructure of the CLC polymer networks could also be controlled by the photopolymerization conditions to improve the sensing performance.

The standard UV curing intensity used in this work was 20 mW/cm^2^. To study the impact of UV curing intensity on the sensing performance of the CLC film, a reduced intensity of 5 mW/cm^2^ was applied to cure the CLC film, and the transient spectral response to 40% ethanol solution was measured as shown in Figure 7a. Comparison with the 20 mW/cm^2^ intensity results presented in Figure 3a shows that reduced UV intensity gave very similar overall behavior. To examine the differences, Figure 7b plots the time-dependent wavelength shifts of CLC films realized with UV intensities of 20 and 5 mW/cm^2^. Both the ethanol-induced wavelength shifts rapidly increased with detection time, reaching a maximum value at 20 s and slowly decaying 30 s later. The time-dependent wavelength shift was larger for higher UV cure intensity because the structure of CLC polymer networks realized by higher UV intensity was relatively sparse and had larger voids. This was because a rougher CLC structural morphology with larger voids allowed the alcohol molecules to easily diffuse into the CLC film and promoted a more significant expansion of the CLCP pitch structure, so that a greater wavelength shift response was induced.

Figure 8 shows the dependence of the wavelength shift of CLCP films realized with UV curing intensities of 5 and 20 mW/cm^2^ on ethanol concentration, in which the spectral response was measured at a detection time of 30 s. Linear fits gave sensitivities of 0.95 and 1.35 nm/% for CLCP films cured at intensities of 5 and 20 mW/cm^2^, respectively. It can be observed that the CLC film cured by higher UV intensity had a higher sensitivity, and higher UV intensity could shorten the process time to cure the CLC film. It concludes that sensor sensitivity could be enhanced by employing an optimum CLC network microstructure to facilitate the diffusion of alcohol molecules into the film and the expansion of the helical structure.

In some applications, it is desirable for a sensor to be reusable. To study the reversibility and repeatability of the spectral response to analytes, the thermal treatment was applied to detach the alcohol molecules from CLCP networks. First, the CLCP film was immersed in 15 mL of 40% ethanol solution for ~30 s. Then, to desorb the alcohol molecules, the CLCP film was taken out of the solution and baked at 50 °C for 1 min after blowing off any liquid droplets on the film surface with a nitrogen stream. Repeating the above immersion and baking process, the individual spectrum of the CLC film after baking was measured, as shown in Figure 9. It can be observed that the helical pitch of the CLCP film could nearly be restored to its original state without any significant loss in performance during these five cycles, indicating the response of CLCs to alcohol yields reversible and repeatable optical response changes. The reversible response was presumably due to the release of alcohol molecules from the CLCP film, leading to restoration of the helical structure. Based on the experimental results, CLCP film could rapidly adsorb and desorb alcohol molecules, which demonstrates that a real-time sensor for discriminating ethanol and methanol in aqueous solution in a fast and reversible manner could be constructed.

The extraordinary property of CLCP can be applied to develop visual sensing materials to alcohol molecules using color changes as a detection signal. It provides a new strategy for measuring the concentration of methanol and ethanol in solution. Moreover, CLCP film with high mechanical stability can be simply fabricated by depositing the LC mixture onto a substrate by inkjet printing or doctor blade coating, potentially enabling the fabrication of large areas in roll-to-roll (R2R) processing at low cost. Therefore, the CLCP film is an excellent candidate for the responsive alcohol sensor, due to its characteristics of ease of manufacture, visibility to the naked eye, fast response, high stability, label-free, battery-free and a microliter volume requirement for the bio-sample solution.

## 3. Materials and Methods

### 3.1. Preparation of CLCP Film

To preserve the polymer integrity of the CLC film, the reactive mesogen and chiral dopant used in this study were diacrylate LC242 and LC756, respectively, obtained from BASF Co. Ltd. (Ludwigshafen, Germany) Paliocolor LC242 is a non-alcohol-responsive reactive mesogen. To facilitate responding alcohol molecules, the photopolymerizable alkyloxybenzoic acids 6OBA and 6OBAM (Synthon Chemicals Ind. Ltd., Bitterfeld-Wolfen, Germany) were used. Additionally, to obtain a fast response and enhance the sensitivity of the CLCP networks on the uptake of the analytes, a nonreactive mesogen 5CB (Tokyo Chemical Ind. Ltd., Tokyo, Japan) as a porogen was also added in the CLC mixture to create macroscopic pores in the polymer networks [19,22]. The free radical photoinitiator used was 2,2-dimethopxy-1,2-diphenyl-ethanone (IRG651, CIBA Co. Ltd., Ikeja, Nigeria). In order to make the transmittance spectra of CLCP upon exposure to alcohol solution cover the visible range, the CLCP film was made by formulating a chiral liquid-crystal mixture of LC242, LC756, 5CB, 6OBA, 6OBAM, and IRG651 in a weight ratio of 34.2:4.8:18:21:21:1. All the LC mixtures were dissolved in a tetrahydrofuran (THF) solution (4:6).

To fabricate the CLCP specimen, a polyimide alignment layer (SE-3140, Nissan Chemical Ind. Ltd., Tokyo, Japan) was first coated on a 2.5 × 2.5 cm cleaned glass substrate, cured, and then rubbed with a velvet cloth to induce planar alignment. Forty microliters of CLC mixture was dropped onto the rubbed substrate, heated at 62 °C for 15 min to volatilize the THF solvent, and then a cover glass substrate coated with fluorinated alkyl silane was attached. After cooling the sample to 59 °C, a shearing force was applied in a horizontal direction to obtain a well-aligned CLC film. Subsequently, the sample was photopolymerized by UV irradiation at intensity 20 mW/cm^2^ for 10 min, and the cover glass was detached after completing the photopolymerization process. To generate the porosity required to enhance sensing ability, the CLCP film was heated on a hot plate at 105 °C for 5 h to extract the nonreactive mesogen 5CB after the polymer networks have been formed. Figure 10 shows the scanning electron micrograph (SEM) of the porosity created in the CLC polymer networks after the extraction of 5CB by heat treatment. The size and density of the porosity in the CLC polymer networks can be controlled by the amount of 5CB in the mixture and the photopolymerization condition [28].

Figure 11 shows the transmission spectra of the CLCP film at different stages. The transmission central wavelength and reflected color of the CLCP would reveal its helical pitch. Initially, a CLCP film was obtained with central wavelength around 536 nm. Upon extraction of 5CB by heat treatment, the transmission central wavelength of the CLC film exhibited a blue shift to around 441 nm (17.7% wavelength reduction) caused by shrinkage of the helical pitch. The reduction ratio in the central wavelength of the CLC film was close to the ratio for the nonreactive mesogen 5CB doped in the CLC mixture, confirming that the heat treatment effectively removed 5CB from the CLCP film.

The photopolymerized alkyloxybenzoic acids of 6OBA and 6OBAM stuck to each other physically to constitute a hydrogen bond by means of the carboxylic moieties [19,22,23]. The carboxylic moieties of CLCP activated during breaking of the hydrogen bridges interacted with the hydroxyl group of analytes, so the bond-breaking treatment stimulated the absorption of alcohol molecules having hydrogen bonds. To enable the absorption of alcohol molecules, the process of breaking hydrogen bonds was performed after the extraction of 5CB by soaking the CLCP film in 20 mL of sodium hydroxide solution (0.5 M). During the process of breaking hydrogen bridges, 6OBA and 6OBAM form carboxylic salt (COO^–^Na^+^), and the helical structure of CLCP with broken hydrogen bond swells. As shown in Figure 11, the spectrum of the CLCP film shows a red shift to around 598 nm after the activation in the alkaline solution. This was a consequence of the CLCP helical structure swelling due to the broken hydrogen bonds in the CLCP networks and the accompanying absorption of the alkaline buffer solution. However, when the CLCP salt film was dried on a hot plate at 50 °C for 3 min, the CLCP film showed a blue shift and returned to the original reflected color after the drying process.

Furthermore, the diffusion rate of alcohol into the CLCP film is critically impacted by the morphology of the polymer network microstructure, which is significantly influenced by the UV curing conditions such as UV light intensity [27,28]. To further address the influence of the microstructure of the CLCP networks on the sensing characteristics of CLC film to alcohol molecules, the UV light at a lower intensity of 5 mW/cm^2^ was also applied to construct the CLCP film to study the ability to differentiate alcohol solutions.

### 3.2. Characterization of the CLCP-Based Responsive Alcohol Sensor

To assess the alcohol-responsive ability of the CLCP film, the transmission spectra of CLCP films were measured for different alcohol solutions. Figure 12 shows a schematic diagram of the test system. A halogen lamp and a Spectra Academy SV-2100 spectrophotometer were used as the incident light source and for measuring the optical spectrum of the CLCP, respectively. To quantitatively investigate the impact of alcohol concentration on the response characteristics of CLCP, 40 μL alcohol solutions with different alcohol concentrations and ratios of methanol to ethanol were dropped onto dried CLC polymer salt films, and the transient spectral characteristics of the CLCP were measured in real time.

## 4. Conclusions

The hydrogen-bonded CLCP film could realize rapid detection of alcohol solutions containing mixed ethanol and methanol and could distinguish methanol and ethanol. The experimental results revealed that ethanol and methanol molecules could make the reflected color of CLCs red shift, while ethanol caused a larger red shift. The helical structure of CLCs swelled upon the adsorption of alcohol molecules, causing a red shift of the transmission band in under 30 s. A highly linear relationship was found between wavelength shift and alcohol concentration. The sensing sensitivity of CLCP networks to alcohol molecules increased as the alcohol concentration increased and the methanol–ethanol ratio decreased. The sensitivities of CLCP networks were 1.35 and 0.18 nm/% to ethanol and methanol, respectively. It was also demonstrated that CLCP film cured with high light intensity had a better optical response to alcohol solution than that cured by low light intensity. Furthermore, it was verified that CLCP film could rapidly absorb and release alcohol, leading to reversible and repeatable optical response changes appropriate for reusable sensor designs. As a result, the CLCP film offers the basis for stable, fast, sensitive, and reversible sensors, which are highly selective due to the differentiated adsorption of alcohols into the micropores of the CLC polymer salt. It is believed that CLCP film holds great potential for use in biosensor applications because of its advantages of linear response and visible color.

## Figures and Tables

**Figure 1 molecules-27-00773-f001:**
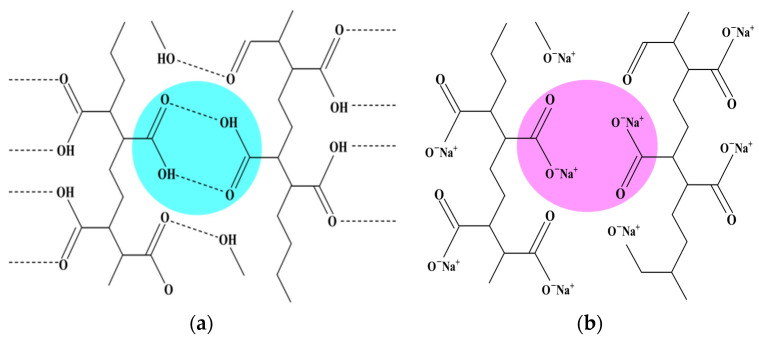
Schematic of (**a**) the dimer formed between 6OBA and 6OBAM through the hydrogen bonds; (**b**) the hydrogen bond was broken by NaOH, and then sodium carboxylate (COO^−^Na^+^) formed.

**Figure 2 molecules-27-00773-f002:**
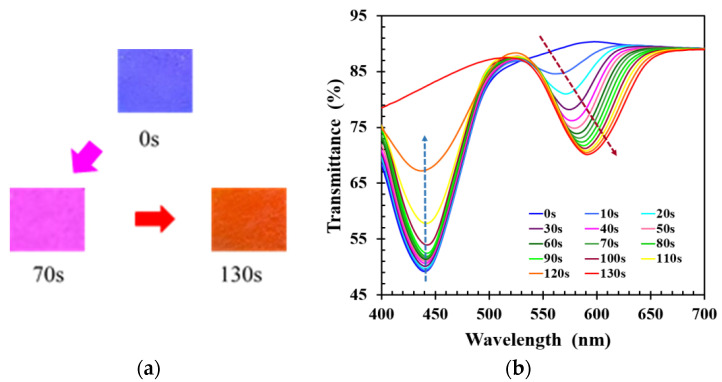
(**a**) The change of the color reflected by CLC film, and (**b**) the transient transmission spectra of CLC film at different immersion times during the process of hydrogen bond breaking.

**Figure 3 molecules-27-00773-f003:**
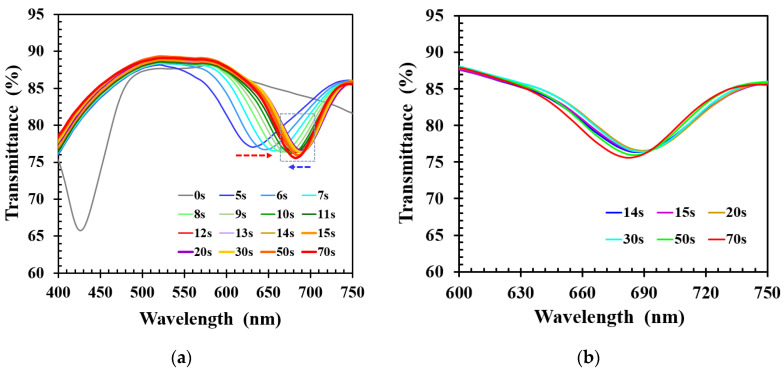
(**a**) Time lapse transmittance spectra of CLC film subjected to 40% ethanol solution, and (**b**) scale up of the blue-shifted region indicated by the blue arrow in (**a**).

**Figure 4 molecules-27-00773-f004:**
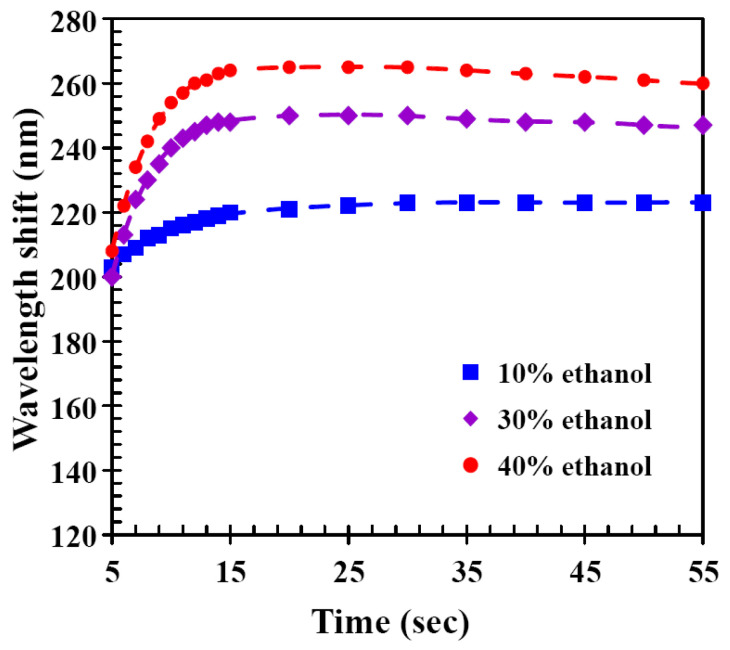
Time-dependent wavelength shifts of CLC film upon exposure to solutions with different ethanol concentrations.

**Figure 5 molecules-27-00773-f005:**
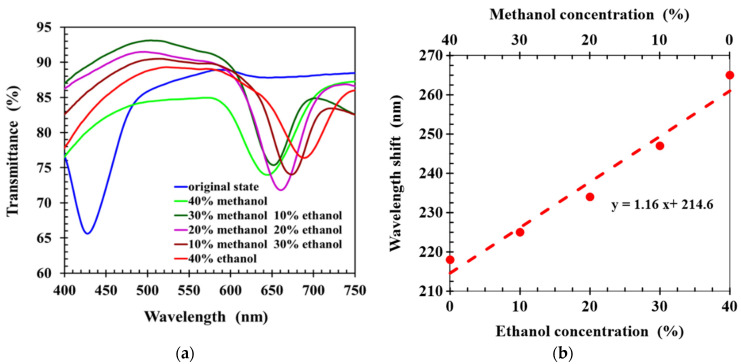
(**a**) The transmission spectra of CLC films, (**b**) wavelength shift of CLCs induced by alcohol solutions (40%) with different ethanol–methanol ratios.

**Figure 6 molecules-27-00773-f006:**
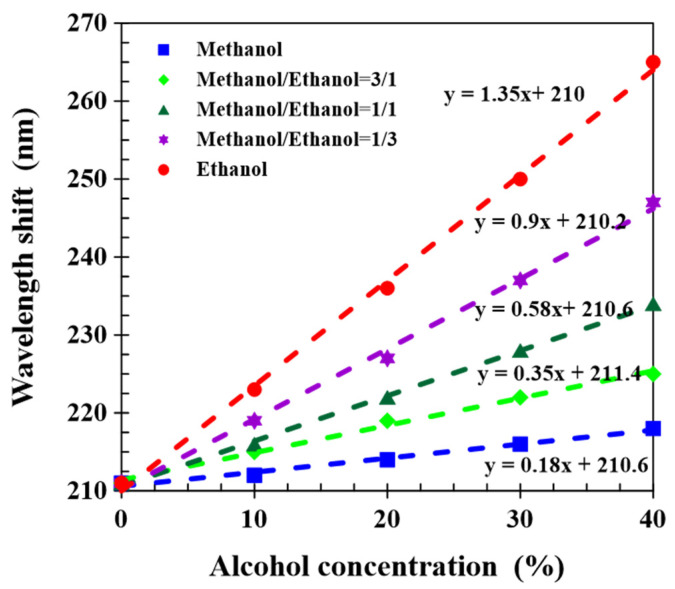
The wavelength shift of CLCP films induced by alcohol solutions, with different total alcohol concentrations and methanol–ethanol ratios.

**Figure 7 molecules-27-00773-f007:**
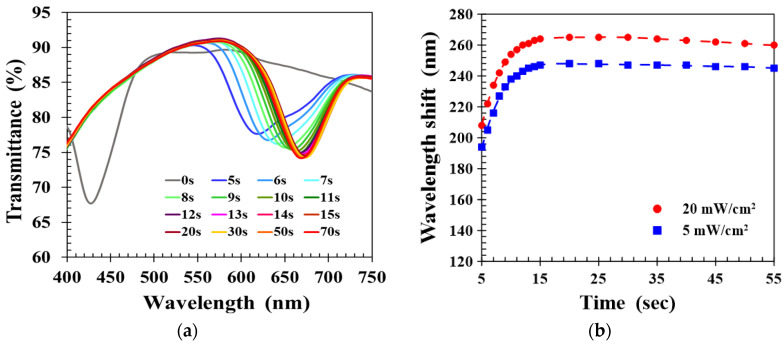
(**a**) Time lapse transmission spectra of CLC film fabricated by UV intensity of 5 mW/cm^2^, and (**b**) comparison of time-dependent wavelength shifts of CLC films realized by different UV intensities.

**Figure 8 molecules-27-00773-f008:**
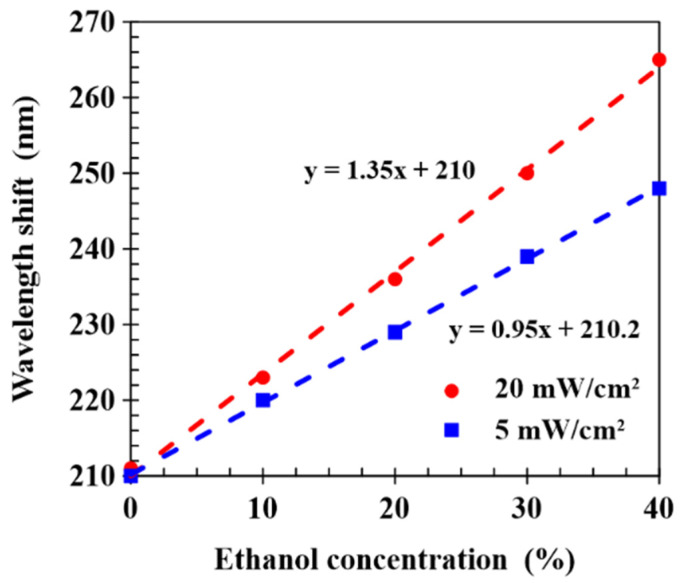
Wavelength shifts of CLCP films cured by different UV intensities and exposed to ethanol concentrations from 0 to 40%.

**Figure 9 molecules-27-00773-f009:**
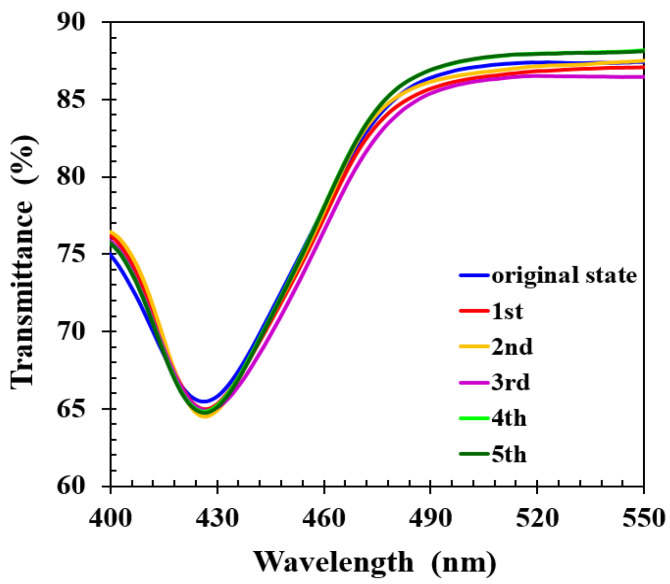
The transmittance spectra of CLC after repeated cycles of immersion in alcohol solution followed by baking.

**Figure 10 molecules-27-00773-f010:**
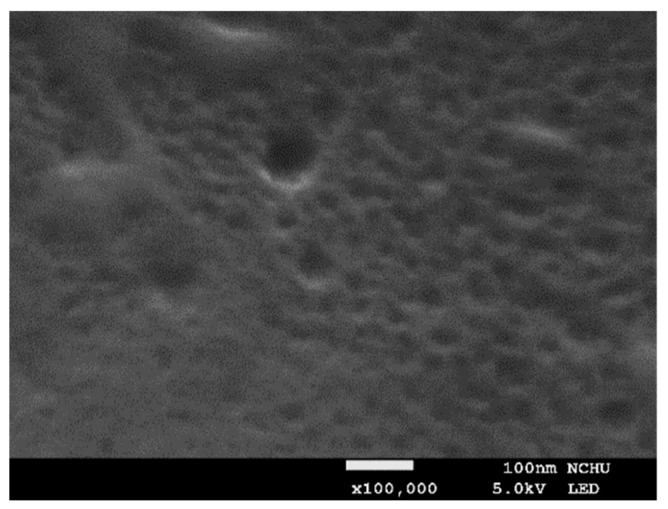
The SEM image is a cross-sectional micrograph of the CLCP film after the extraction of 5CB.

**Figure 11 molecules-27-00773-f011:**
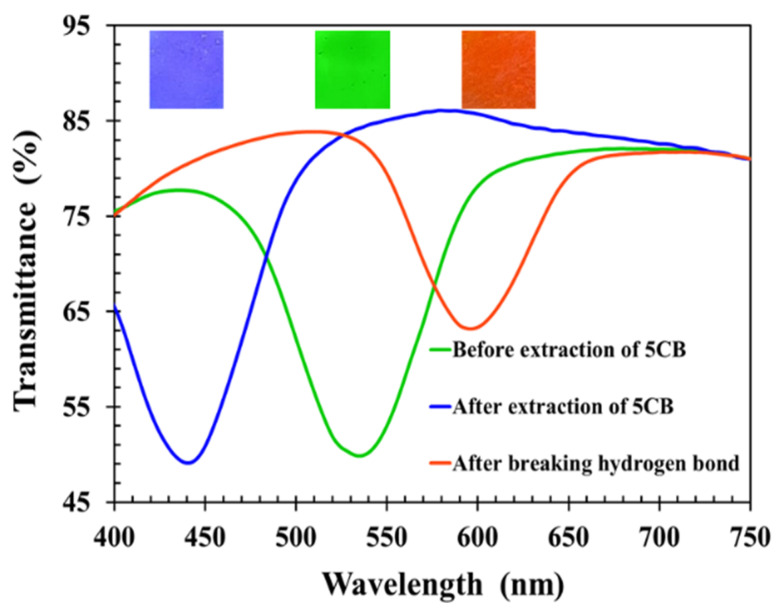
Transmission spectra of CLC films at different process steps.

**Figure 12 molecules-27-00773-f012:**
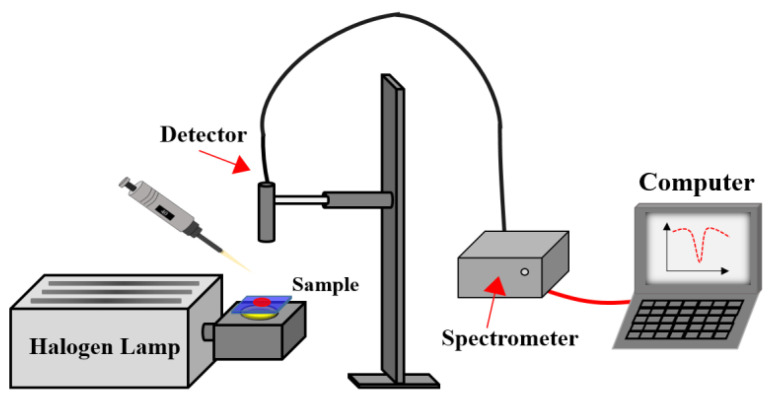
Schematic diagram of the system for evaluating the sensing properties of CLC film.

## Data Availability

No new data were created or analyzed in this study. Data sharing is not applicable to this article.

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
