# Peer review of "Alcohol Selective Optical Sensor Based on Porous Cholesteric Liquid Crystal Polymer Networks"

_molecules, 2022, doi:10.3390/molecules27030773_

Round 1

Reviewer 1 Report

In this study, a linear relationship between wavelength shift of CLCP networks and alcohol concentration was obtained experimentally and the sensor characteristics were explored. The study is recommended for publication after following comments are being addressed successfully.

  1. English language of manuscript needs some improvement.
  2. Please avoid the word ‘we’ throughout the manuscript.
  3. Abstract of manuscript needs significant improvement. Please include the important findings with some numerical values. Similarly, conclusion of manuscript needs significant improvement.
  4. Gap of knowledge and novelty of study need to be clearer in the Introduction part of manuscript.
  5. Results must be discussed more in depth, please improve the discussion.
  6. Results and discussion need to be compared with previous published studies and supported by references.

Reviewer 2 Report

The manuscript by Yeh et al. proposed fabrication of cholesteric liquid crystal polymer networks for sensing the mixture of alcohol solutions. A highly responsive alcohol optical sensor was realized by make porous sensing film. The work is interesting. The study can be accepted after some solving the following issues.

  • The full name of 6OBA and 6OBAM should be given when first appeared in the manuscript; the chemical structure change to show the mechanism in Fig. 9 is better put ahead as Fig.1 to assist the reader clearly understand the detection principle.
  • The author fabricated the porous sensing film, the SEM image of the film should be provided.
  • The alcohol will swell the film to induce the wavelength shifting. What about the swelling degree of the porous film in different alcohol solutions?
  • The author tested the optical spectra change of sensing film with the alcohol solutions up to 40%, what about the higher concentration?
  • What is the sensors’ performance when varying the concentration from high to low?
  • The author discovered that the UV curing intensity will influence the sensitivity of the sensor and compared the sensor fabricated with 5 and 20 mW/cm2 UV intensity. I concerned about the performance of sensor fabricated with higher UV intensity.
  • The power unit in Fig. 6b: ‘2’ should be put as superscript.

Reviewer 3 Report

The submitted manuscript describes a responsive cholesteric liquid crystal polymer (CLCP) film with controlled porosity as an optical sensor to distinguish between methanol and ethanol in alcoholic solutions. In order to facilitate responding to the alcohols, porosity is generated by removing non-reactive liquid crystal agents and the hydrogen bridges of CLCP are broken. Alterations in the central wavelength of the CLCP network transmission spectrum allow the methanol-ethanol ratio to be discriminated. Moreover, the impact of UV intensity for curing CLC mixture on the sensing sensitivity to the different alcohol concentrations and ratios of methanol/ethanol was also studied. The higher UV intensity can enhance the sensitivity to alcohol molecules and distinguishing ability between methanol and ethanol. The CLCP sensor provides an easy-to-use, low-cost, rapid, and repeatable method for the detection and quantification of methanol contamination in ethanol solutions and shows strong potential as a cost-effective sensor for other 18 liquid sensing applications. This research is well arranged, has a sequence of clear ideas, and concise writing that fits the research plan and methodology. The literature review is good and they were able to successfully discuss a discussion of their progress, from both a perspective and an applied perspective. The method they choose makes this data analysis excellent research and enables them to answer research questions and test their hypotheses. Hereby, I recommend this manuscript for publication in molecules after minor revision.

Introduction:

  • Insert a new paragraph to explain the advantages of liquid crystal polymer (CLCP) with respect to other utilizing methods?
  • Clarify the benefits of porous Polymers make it the desired choice than others?

Results and discussion

  • Is the optical term mainly refer to the fluorescence technique?

I suggest using a colorimetric sensor term instead

  • The advantages of this method in comparison with other methods should be highlighted, including analytical characteristics, reproducibility, specificity, stability?
  • The validation of this technique should be introduced by comparison with a previously validated method?

Round 2

Reviewer 1 Report

The manuscript is now recommended for publication in Molecules journal.